# SubmodBoxes: Near-Optimal Search for a Set of Diverse Object Proposals

**Qing Sun**
Virginia Tech
sunqing@vt.edu

Dhruv Batra
Virginia Tech
https://mlp.ece.vt.edu/

## Abstract

This paper formulates the search for a set of bounding boxes (as needed in object proposal generation) as a monotone submodular maximization problem over the space of *all possible bounding boxes in an image*. Since the number of possible bounding boxes in an image is very large $O(\#pixels^2)$, even a single linear scan to perform the greedy augmentation for submodular maximization is intractable. Thus, we formulate the greedy augmentation step as a Branch-and-Bound scheme. In order to speed up repeated application of B&B, we propose a novel generalization of Minoux's 'lazy greedy' algorithm to the B&B tree. Theoretically, our proposed formulation provides a new understanding to the problem, and contains classic heuristic approaches such as Sliding Window+Non-Maximal Suppression (NMS) and and Efficient Subwindow Search (ESS) as special cases. Empirically, we show that our approach leads to a state-of-art performance on object proposal generation via a novel diversity measure.

## 1 Introduction

A number of problems in Computer Vision and Machine Learning involve searching for a set of bounding boxes or rectangular windows. For instance, in object detection [9, 16, 17, 19, 34, 36, 37], the goal is to output a set of bounding boxes localizing all instances of a *particular* object category. In object proposal generation [2, 7, 39, 41], the goal is to output a set of candidate bounding boxes that may potentially contain an object (of *any* category). Other scenarios include face detection, multi-object tracking and weakly supervised learning [10].

**Classical Approach: Enumeration + Diverse Subset Selection.** In the context of object detection, the classical paradigm for searching for a set of bounding boxes used to be:

- **Sliding Window** [9, 16, 40]: *i.e.*, enumeration over all windows in an image with some level of sub-sampling, followed by
- **Non-Maximal Suppression (NMS)**: *i.e.*, picking a spatially-diverse set of windows by suppressing windows that are too close or overlapping.

As several previous works [3, 26, 40] have recognized, the problem with this approach is inefficiency – the number of possible bounding boxes or rectangular subwindows in an image is $O(\#pixels^2)$. Even a low-resolution (320 x 240) image contains more than *one billion* rectangular windows [26]!

As a result, modern object detection pipelines [17, 19, 36] often rely on object proposals as a pre-processing step to reduce the number of candidate object locations to a few hundreds or thousands (rather than billions).

Interestingly, this migration to object proposals has simply *pushed the problem (of searching for a set of bounding boxes) upstream*. Specifically, a number of object proposal techniques [8, 32, 41] involve the same enumeration + NMS approach – except they typically use cheaper features to be a fast proposal generation step.

**Goal.** The goal of this paper is to formally study the search for a set of bounding boxes as an optimization problem. Clearly, enumeration + post-processing for diversity (via NMS) is one widely-used heuristic approach. Our goal is to formulate a formal optimization objective and propose an efficient algorithm, ideally with guarantees on optimization performance.

**Challenge.** The key challenge is the exponentially-large search space – the number of possible $M$-sized sets of bounding boxes is $\binom{O(\#pixels^2)}{M} = O(\#pixels^{2M})$ (assuming $M \leq \#pixels^2/2$).

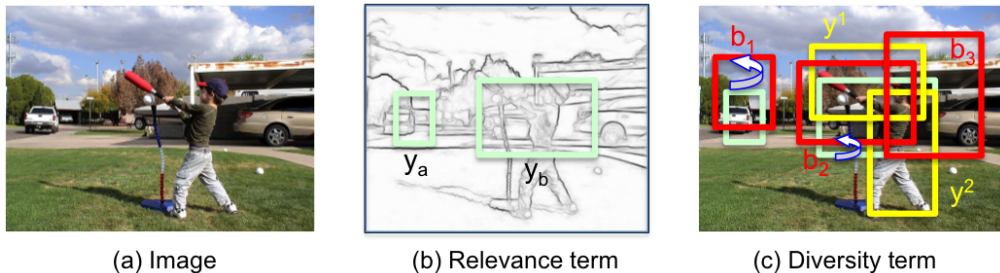

| (a) Image | (b) Relevance term | (c) Diversity term |

Figure 1: Overview of our formulation: SubmodBoxes. We formulate the selection of a set of boxes as a constrained submodular maximization problem. The objective and marginal gains consists of two parts: relevance and diversity. Figure (b) shows two candidate windows $\mathbf{y}_a$ and $\mathbf{y}_b$. Relevance is the sum of edge strength over all edge groups (black curves) wholly enclosed in the window. Figure (c) shows the diversity term. The marginal gain in diversity due to a new window ($\mathbf{y}_a$ or $\mathbf{y}_b$) is the ability of the new window to *cover* the reference boxes that are currently not well-covered with the already chosen set $Y = \{\mathbf{y}^1, \mathbf{y}^2\}$. In this case, we can see that $\mathbf{y}_a$ covers a new reference box $b_1$. Thus, the marginal gain in diversity of $\mathbf{y}_a$ will be larger than $\mathbf{y}_b$.

**Overview of our formulation: SubmodBoxes.** Let $\mathcal{Y}$ denote the set of all possible bounding boxes or rectangular subwindows in an image. This is a structured output space [4,21,38], with the size of this set growing quadratically with the size of the input image, $|\mathcal{Y}| = O(\#pixels^2)$.

We formulate the selection of a set of boxes as a search problem on the power set $2^{\mathcal{Y}}$. Specifically, given a budget of $M$ windows, we search for a set $Y$ of windows that are both *relevant* (*e.g.*, have high likelihood of containing an object) and *diverse* (to cover as many objects instances as possible):

$$\underbrace{\underset{Y \in 2^{\mathcal{Y}}}{\text{argmax}}}_{\text{search over power-set}} \quad \underbrace{F(Y)}_{\text{objective}} = \underbrace{R(Y)}_{\text{relevance}} + \underbrace{\lambda}_{\text{trade-off parameter}} \underbrace{D(Y)}_{\text{diversity}} \quad s.t. \quad \underbrace{|Y| \leq M}_{\text{budget constraint}} \qquad (1)$$

Crucially, when the objective function $F : 2^{\mathcal{Y}} \to \mathbb{R}$ is *monotone* and *submodular*, then a simple greedy algorithm (that iteratively adds the window with the largest *marginal gain* [24]) achieves a near-optimal approximation factor of $(1 - \frac{1}{e})$ [24,30].

Unfortunately, although conceptually simple, this greedy augmentation step requires an enumeration over the space of all windows $\mathcal{Y}$ and thus a naïve implementation is intractable.

In this work, we show that for a broad class of relevance and diversity functions, this greedy augmentation step may be efficiently formulated as a Branch-and-Bound (B&B) step [12,26], with easily computable upper-bounds. This enables an efficient implementation of greedy, with significantly fewer evaluations than a linear scan over $\mathcal{Y}$.

Finally, in order to speed up repeated application of B&B across iterations of the greedy algorithm, we present a novel generalization of Minoux's 'lazy greedy' algorithm [29] to the B&B tree, where different branches are explored in a lazy manner in each iteration.

We apply our proposed technique SubmodBoxes to the task of generating object proposals [2,7,39,41] on the PASCAL VOC 2007 [13], PASCAL VOC 2012 [14], and MS COCO [28] datasets. Our results show that our approach outperforms all baselines.

**Contributions.** This paper makes the following contributions:

1. We formulate the search for a set of bounding boxes or subwindows as the constrained maximization of a monotone submodular function. To the best of our knowledge, despite the popularity of object recognition and object proposal generation, this is the first such formal optimization treatment of the problem.

2. Our proposed formulation contains existing heuristics *as special cases*. Specifically, Sliding Window + NMS can be viewed as an instantiation of our approach under a specific definition of the diversity function $D(\cdot)$.

3. Our work can be viewed as a generalization of the 'Efficient Subwindow Search (ESS)' of Lampert *et al.* [26], who proposed a B&B scheme for finding the *single* best bounding box in an image. Their extension to detecting multiple objects consisted of a heuristic for 'suppressing' features extracted from the selected bounding box and re-running the procedure. We show that this heuristic is a special case of our formulation under a specific diversity function, thus providing theoretical justification to their intuitive heuristic.

4. To the best of our knowledge, our work presents the first generalization of Minoux's 'lazy greedy' algorithm [29] to structured-output spaces (the space of bounding boxes).

5. Finally, our experimental contribution is a novel diversity measure which leads to state-of-art performance on the task of generating object proposals.

## 2 Related Work

Our work is related to a few different themes of research in Computer Vision and Machine Learning.

**Submodular Maximization and Diversity.** The task of searching for a diverse high-quality subset of items from a ground set has been well-studied in a number of application domains [6, 11, 22, 25, 27, 31], and across these domains submodularity has emerged as an a fundamental property of set functions for measuring diversity of a subset of items. Most previous work has focussed on submodular maximization on *unstructured spaces*, where the ground set is efficiently enumerable.

Our work is closest in spirit to Prasad *et al*. [31], who studied submodular maximization on *structured* output spaces, *i.e.* where each item in the ground set is itself a structured object (such as a segmentation of an image). Unlike [31], our ground set $\mathcal{Y}$ is not exponentially large, only 'quadratically' large. However, enumeration over the ground set for the greedy-augmentation step is still infeasible, and thus we use B&B. Such structured output spaces and greedy-augmentation oracles were not explored in [31].

**Bounding Box Search in Object Detection and Object Proposals.** As we mention in the introduction, the search for a set of bounding boxes via heuristics such as Sliding Window + NMS used to be the dominant paradigm in object recognition [9, 16, 40]. Modern pipelines have shifted that search step to object proposal algorithms [17, 19, 36]. A comparison and overview of object proposals may be found in [20]. Zitnick *et al*. [41] generate candidate bounding boxes via Sliding Window + NMS based on an "objectness" score, which is a function of the number of contours wholly enclosed by a bounding box. We use this objectness score as our relevance term, thus making SubmodBoxes directly comparable to NMS. Another closely related work is [18], which presents an 'active search' strategy for reranking selective search [39] object proposals based on a contextual cues. Unlike this work, our formulation is not restricted to any pre-selected set of windows. We search over the entire power set $2^{\mathcal{Y}}$, and may generate any possible set of windows (up to convergence tolerance in B&B).

**Branch-and-Bound.** One key building block of our work is the 'Efficient Subwindow Search (ESS)' B&B scheme *et al*. [26]. ESS was originally proposed for single-instance object detection. Their extension to detecting multiple objects consisted of a heuristic for 'suppressing' features extracted from the selected bounding box and re-running the procedure. In this work, we extend and generalize ESS in multiple ways. First, we show that relevance (objectness scores) and diversity functions used in object proposal literature are amenable to upper-bound and thus B&B optimization. We also show that the 'suppression' heuristic used by [26] is a special case of our formulation under a specific diversity function, thus providing theoretical justification to their intuitive heuristic. Finally, [3] also proposed the use of B&B for NMS in object detection. Unfortunately, as we explain later in the paper, the NMS objective is submodular *but not monotone*, and the classical greedy algorithm does not have approximation guarantees in this setting. In contrast, our work presents a general framework for bounding-box subset-selection based on *monotone* submodular maximization.

## 3 SubmodBoxes: Formulation and Approach

We begin by establishing the notation used in the paper.

**Preliminaries and Notation.** For an input image $\mathbf{x}$, let $\mathcal{Y}_{\mathbf{x}}$ denote the set of all possible bounding boxes or rectangular subwindows in this image. For simplicity, we drop the explicit dependance on $\mathbf{x}$, and just use $\mathcal{Y}$. Uppercase letters refer to set functions $F(\cdot), R(\cdot), D(\cdot)$, and lowercase letter refer to functions over individual items $f(\mathbf{y}), r(\mathbf{y})$.

A set function $F : 2^{\mathcal{Y}} \to \mathbb{R}$ is submodular if its *marginal gains* $F(b|S) \equiv F(S \cup b) - F(S)$ are decreasing, *i.e.* $F(b|S) \geq F(b|T)$ for all sets $S \subseteq T \subseteq \mathcal{Y}$ and items $b \notin T$. The function $F$ is called *monotone* if adding an item to a set does not hurt, *i.e.* $F(S) \leq F(T), \ \forall S \subseteq T$.

**Constrained Submodular Maximization.** From the classical result of Nemhauser [30], it is known that cardinality constrained maximization of a monotone submodular $F$ can be performed near-optimally via a greedy algorithm. We start out with an empty set $Y^0 = \emptyset$, and iteratively add the next 'best' item with the largest marginal gain over the chosen set :

$$Y^t = Y^{t-1} \cup y^t, \qquad \text{where} \qquad y^t = \underset{y \in \mathcal{Y}}{\operatorname{argmax}} \ F(y \mid Y^{t-1}). \qquad (2)$$

The score of the final solution $Y^M$ is within a factor of $(1 - \frac{1}{e})$ of the optimal solution. The computational bottleneck is that in each iteration, we must find the item with the largest marginal gain. In our case, $|\mathcal{Y}|$ is the space of all rectangular windows in an image, and exhaustive enumeration

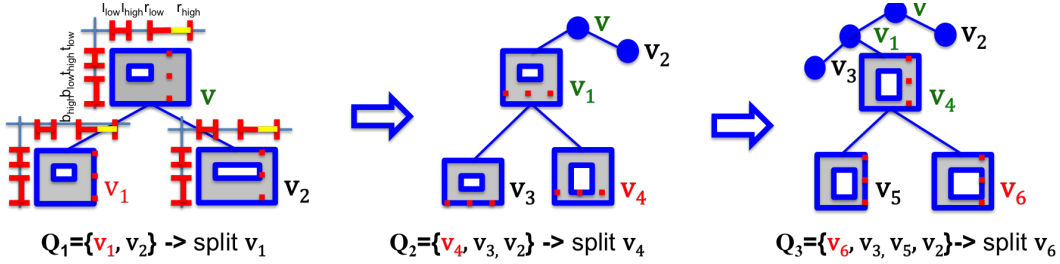

**Figure 2:** Priority queue in B&B scheme. Each vertex in the tree represents a set of windows. Blue rectangles denote the largest and the smallest window in the set. Gray region denotes the rectangle set $\mathcal{Y}_v$. In each case, the priority queue consists of all leaves in the B&B tree ranked by the upper bound $U_v$. *Left:* shows vertex $v$ is split along the *right* coordinate interval into equal halves: $v_1$ and $v_2$. *Middle:* The highest priority vertex $v_1$ in $Q_1$ is further split along *bottom* coordinate into $v_3$ and $v_4$. *Right:* The highest priority vertex $v_4$ in $Q_2$ is split along *right* coordinate into $v_5$ and $v_6$. This procedure is repeated until the highest priority vertex in the queue is a single rectangle.

is intractable. Instead of exploring subsampling as is done in Sliding Window methods, we will formulate this greedy augmentation step as an optimization problem solved with B&B.

**Sets vs Lists.** For pedagogical reasons, our problem setup is motivated with the language of sets $(\mathcal{Y}, 2^{\mathcal{Y}})$ and subsets $(Y)$. In practice, our work falls under submodular *list* prediction [11, 33, 35]. The generalization from sets to lists allows reasoning about an ordering of the items chosen and (potentially) repeated entries in the list. Our final solution $Y^M$ is an (ordered) list not an (unordered) set. All guarantees of greedy remain the same in this generalization [11, 33, 35].

### 3.1 Parameterization of $\mathcal{Y}$ and Branch-and-Bound Search

In this subsection, we briefly recap the Efficient Subwindow Search (ESS) of Lampert *et al.* [26], which is used a key building block in this work. The goal of [26] is to maximize a (potentially non-smooth) objective function over the space of all rectangular windows: $\max_{\mathbf{y} \in \mathcal{Y}} f(\mathbf{y})$.

A rectangular window $\mathbf{y} \in \mathcal{Y}$ is parameterized by its top, bottom, left, and right coordinates $\mathbf{y} = (t, b, l, r)$. A *set* of windows is represented by using intervals for each coordinate instead of a single integer, for example $[T, B, L, R]$, where $T = [t_{low}, t_{high}]$ is a range. In this parameterization, the set of all possible boxes in an $(h \times w)$-sized image can be written as $\mathcal{Y} = [[1, h], [1, h], [1, w], [1, w]]$.

**Branch-and-Bound over $\mathcal{Y}$.** ESS creates a B&B tree, where each vertex $v$ in the tree is a rectangle set $\mathcal{Y}_v$ and an associated upper-bound on the objective function achievable in this set, *i.e.* $\max_{\mathbf{y} \in \mathcal{Y}_v} f(\mathbf{y}) \leq U_v$. Initially, this tree consists of a single vertex, which is the entire search space $\mathcal{Y}$ and (typically) a loose upper-bound. ESS proceeds in a best-first manner [26]. In each iteration, the vertex/set with the highest upper-bound is chosen for branching, and then new upper-bounds are computed on each of the two children/sub-sets created. In practice, this is implemented with a priority queue over the vertices/sets that are currently leaves in the tree. Fig. 2 shows an illustration of this procedure. The parent rectangle set is split along its largest coordinate interval into two equal halves, thus forming disjoint children sets. B&B explores the tree in a best-first manner till a single rectangle is identified with a score *equal* to the upper-bound at which point we have found a global optimum. In our experiments, we show results with different convergence tolerances.

**Objective.** In our setup, the objective (at each greedy-augmentation step) is the marginal gain of the window $\mathbf{y}$ w.r.t. the currently chosen list of windows $Y^{t-1}$, *i.e.* $f(\mathbf{y}) = F(\mathbf{y} \mid Y^{t-1}) = R(\mathbf{y} \mid Y^{t-1}) + \lambda D(\mathbf{y} \mid Y^{t-1})$. In the following subsections, we describe the relevance and diversity terms in detail, and show how upper bounds can be efficiently computed over the sets of windows.

### 3.2 Relevance Function and Upper Bound

The goal of the relevance function $R(Y)$ is to quantify the "quality" or "relevance" of the windows chosen in $Y$. In our work, we define $R(Y)$ to be a *modular* function aggregating the quality of all chosen windows *i.e.* $R(Y) = \sum_{\mathbf{y} \in Y} r(\mathbf{y})$. Thus, the marginal gain of window $\mathbf{y}$ is simply its individual quality regardless of what else has already been chosen, *i.e.* $R(\mathbf{y} \mid Y^{t-1}) = r(\mathbf{y})$.

In our application of object proposal generation, we use the objectness score produced by Edge-Boxes [41] as our relevance function. The main intuition of EdgeBoxes is that the number of contours or "edge groups" wholly contained in a box is indicative of its objectness score. Thus, it first creates a grouping of edge pixels called edge groups, each associated with a real-valued edge strength $s_i$.

Abstracting away some of the domain-specific details, EdgeBoxes essentially defines the score of a box as a weighted sum of the strengths of edge groups contained in it, normalized by the size of the

box *i.e.* EdgeBoxesScore$(\mathbf{y}) = \dfrac{\sum_{\text{edge group } i \in \mathbf{y}} w_i s_i}{\text{size-normalization}}$, where with a slight abuse of notation, we use 'edge group $i \in \mathbf{y}$' to mean the edge groups contained the rectangle $\mathbf{y}$.

These weights and size normalizations were found to improve performance of EdgeBoxes. In our work, we use a simplification of the EdgeBoxesScore which allow for easy computation of upper bounds:

$$r(\mathbf{y}) = \frac{\sum_{\text{edge group } i \in \mathbf{y}} s_i}{\text{size-normalization}}, \tag{3}$$

*i.e.*, we ignore the weights. One simple upper-bound for a set of windows $\mathcal{Y}_v$ can be computed by accumulating *all possible* positive scores and *the least necessary* negative scores:

$$\max_{\mathbf{y} \in \mathcal{Y}_v} r(\mathbf{y}) \leq \frac{\sum_{\text{edge group } i \in \mathbf{y}_{\max}} s_i \cdot [\![ s_i \geq 0 ]\!] + \sum_{\text{edge group } i \in \mathbf{y}_{\min}} s_i \cdot [\![ s_i \leq 0 ]\!]}{\text{size-normalization}(\mathbf{y}_{\min})}, \tag{4}$$

where $\mathbf{y}_{\max}$ is the largest and $\mathbf{y}_{\min}$ is the smallest box in the set $\mathcal{Y}_v$; and $[\![ \cdot ]\!]$ is the Iverson bracket.

Consistent with the experiments in [41], we found that this simplification indeed hurts performance in the EdgeBoxes Sliding Window + NMS pipeline. However, interestingly we found that even with this weaker relevance term, SubmodBoxes was able to outperform EdgeBoxes. Thus, the drop in performance due to a weaker relevance term was more than compensated for by the ability to perform B&B jointly on the relevance and diversity terms.

### 3.3 Diversity Function and Upper Bound

The goal of the diversity function $D(Y)$ is to encourage non-redundancy in the chosen set of windows and potentially capture different objects in the image. Before we introduce our own diversity function, we show how existing heuristics in object detection and proposal generation can be written as special cases of this formulation, under specific diversity functions.

**Sliding Window + NMS.** Non-Maximal Suppression (NMS) is the most popular heuristic for selecting diverse boxes in computer vision. NMS is typically explained *procedurally* – select the highest scoring window $\mathbf{y}^1$, suppress all windows that overlap with $\mathbf{y}^1$ by more than some threshold, select the next highest scoring window $\mathbf{y}^2$, rinse and repeat.

This procedure can be explained as a special case of our formulation. Sliding Window corresponds to enumeration over $\mathcal{Y}$ with some level of sub-sampling (or stride), typically with a fixed aspect ratio. Each step in NMS is precisely a greedy augmentation step under the following marginal gain:

$$\underset{\mathbf{y} \in \mathcal{Y}_{\text{sub-sampled}}}{\operatorname{argmax}} \; r(\mathbf{y}) + \lambda D_{NMS}(\mathbf{y} \mid Y^{t-1}), \quad \text{where} \tag{5a}$$

$$D_{NMS}(\mathbf{y} \mid Y^{t-1}) = \begin{cases} 0 & \text{if} \quad \max_{\mathbf{y}' \in Y^{t-1}} \text{IoU}(\mathbf{y}', \mathbf{y}) \leq \text{NMS-threshold} \\ -\infty & \text{else.} \end{cases} \tag{5b}$$

Intuitively, the NMS diversity function imposes an infinite penalty if a new window $\mathbf{y}$ overlaps with a previously chosen $\mathbf{y}'$ by more than a threshold, and offers no reward for diversity beyond that. This explains the NMS procedure of suppressing overlapping windows and picking the highest scoring one among the unsuppressed ones. Notice that this diversity function is submodular but *not monotone* (the marginals gains may be negative). A similar observation was made in [3]. For such non-monotone functions, greedy does not have approximation guarantees and different techniques are needed [5, 15]. This is an interesting perspective on the classical NMS heuristic.

**ESS Heuristic [26].** ESS was originally proposed for single-instance object detection. Their extension to detecting multiple instances consisted of a heuristic for suppressing the features extracted from the selected bounding box and re-running the procedure. Since their scoring function was linear in the features, this heuristic of suppressing features and rerunning B&B can be expressed as a greedy augmentation step under the following marginal gain:

$$\underset{\mathbf{y} \in \mathcal{Y}}{\operatorname{argmax}} \; r(\mathbf{y}) + \lambda D_{ESS}(\mathbf{y} \mid Y^{t-1}), \text{ where } D_{ESS}(\mathbf{y} \mid Y^{t-1}) = -r\left(\mathbf{y} \cap (\mathbf{y}^1 \cup \mathbf{y}^2 \ldots \mathbf{y}^{t-1})\right) \tag{6}$$

*i.e.*, the ESS diversity function *subtracts* the score contribution coming from the intersection region. If the $r(\cdot)$ is non-negative, it is easy to see that this diversity function is monotone and submodular – adding a new window never hurts, and since the marginal gain is the score contribution of the *new regions* not covered by previous window, it is naturally diminishing. Thus, even though this heuristic not was presented as such, the authors of [26] did in fact formulate a near-optimal greedy algorithm for maximizing a monotone submodular function. Unfortunately, while $r(\cdot)$ is always positive in our experiments, this was not the case in the experimental setup of [26].

**Our Diversity Function.** Instead of hand-designing an explicit diversity function, we use a function that implicitly measures diversity in terms of *coverage* of a set of *reference* set of bounding boxes $B$. This reference set of boxes may be a uniform sub-sampling of the space of windows as done in Sliding Window methods, or may itself be the output of another object proposal method such as Selective Search [39]. Specifically, each greedy augmentation step under our formulation given by:

$$\underset{\mathbf{y}\in\mathcal{Y}}{\operatorname{argmax}} \; r(\mathbf{y}) + \lambda D_{\text{coverage}}(\mathbf{y} \mid Y^{t-1}), \; \text{where} \; D_{\text{coverage}}(\mathbf{y} \mid Y^{t-1}) = \max_{b\in B} \delta\text{IoU}(\mathbf{y}, b \mid Y^{t-1}) \quad (7a)$$

$$\delta\text{IoU}(\mathbf{y}, b \mid Y^{t-1}) = \max\{\text{IoU}(\mathbf{y}, b) - \max_{\mathbf{y}'\in Y^{t-1}} \text{IoU}(\mathbf{y}', b), 0\}. \quad (7b)$$

Intuitively speaking, the marginal gain of a new window $\mathbf{y}$ under our diversity function is the largest gain in coverage of exactly one of the references boxes. We can also formulate this diversity function as a maximum bipartite matching problem between the reference proposal boxes $Y$ and the reference boxes $B$ (in our experiments, we also study performance under top-k matches). We show in the supplement that this marginal gain is always non-negative and decreasing with larger $Y^{t-1}$, thus the diversity function is monotone submodular. All that remains is to compute an upper-bound on this marginal gain. Ignoring constants, the key term to bound is $\text{IoU}(\mathbf{y}, b)$. We can upper-bound this term by computing the intersection w.r.t. the largest window in the window set $\mathbf{y}_{\max}$, and computing the union w.r.t. to the smallest window $\mathbf{y}_{\min}$, *i.e.* $\max_{\mathbf{y}\in\mathcal{Y}_v} \text{IoU}(\mathbf{y}, b) \leq \frac{area(\mathbf{y}_{\max}\cap b)}{area(\mathbf{y}_{\min}\cup b)}$.

## 4 Speeding up Greedy with Minoux's 'Lazy Greedy'

In order to speed up repeated application of B&B across iterations of the greedy algorithm, we now present an application of Minoux's 'lazy greedy' algorithm [29] to the B&B tree.

The key insight of classical lazy greedy is that the marginal gain function $F(\mathbf{y} \mid Y^t)$ is a non-increasing function of $t$ (due to submodularity of $F$). Thus, at time $t-1$, we can *cache* the priority queue of marginals gains $F(\mathbf{y} \mid Y^{t-2})$ for all items. At time $t$, lazy greedy does not recompute all marginal gains. Rather, the item at the front of the priority queue is picked, its marginal gain is updated $F(\mathbf{y} \mid Y^{t-1})$, and the item is reinserted into the queue. Crucially, if the item remains at the front of the priority queue, lazy greedy can stop, and we have found the item with the largest marginal gain.

**Interleaving Lazy Greedy with B&B.** In our work, the priority queue does not contain single items, rather sets of windows $\mathcal{Y}_v$ corresponding to the vertices in the B&B tree. Thus, we must interleave the lazy updates with the Branch-and-Bound steps. Specifically, we pick a set from the front of the queue and compute *the upper-bound* on its marginal gain. We reinsert this set into the priority queue. Once a set remains at the front of the priority queue after reinsertion, we have found the set with the highest upper-bound. This is when perform a B&B step, *i.e.* split this set into two children, compute the upper-bounds on the children, and insert them into the queue.

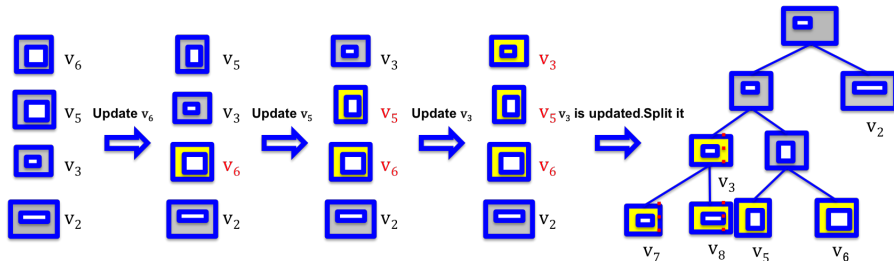

Figure 3: Interleaving Lazy Greedy with B&B. The first few steps update upper-bounds, following by finally branching on a set. Some sets, such as $v_2$ are never updated or split, resulting in a speed-up.

Fig. 3 illustrates how the priority queue and B&B tree are updated in this process. Suppose at the end of iteration $t-1$ and the beginning of iteration $t$, we have the priority queue shown on the left. The first few updates involve recomputing the upper-bounds on the window sets $(v_6, v_5, v_3)$, following by branching on $v_3$ because it continues to stay on top of the queue, creating new vertices $v_7, v_8$. Notice that $v_2$ is never explored (updated or split), resulting in a speed-up.

## 5 Experiments

**Setup.** We evaluate SubmodBoxes for object proposal generation on three datasets: PASCAL VOC 2007 [13], PASCAL VOC 2012 [14], and MS COCO [28]. The goal of experiments is to validate our approach by testing the accuracy of generated object proposals and the ability of handling different kinds of reference boxes, and observe trends as we vary multiple parameters.

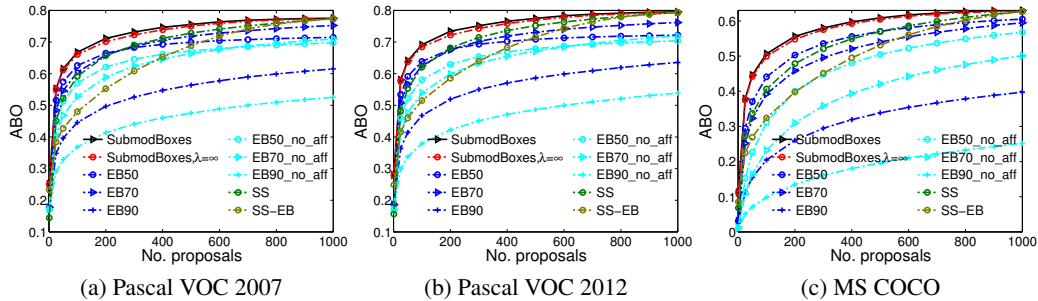

|  |  |  |
|---|---|---|
| (a) Pascal VOC 2007 | (b) Pascal VOC 2012 | (c) MS COCO |

Figure 4: ABO vs. No. Proposals.

**Evaluation.** To evaluate the quality of our object proposals, we use Mean Average Best Overlap (MABO) score. Given a set of ground-truth boxes $\text{GT}^c$ for a class $c$, ABO is calculated by averaging the best IoU between each ground truth bounding box and all object proposals:

$$\text{ABO}^c = \frac{1}{|\text{GT}^c|} \sum_{g \in \text{GT}^c} \max_{\mathbf{y} \in Y} \text{IoU}(g, \mathbf{y}) \tag{8}$$

MABO is a mean ABO over all classes.

**Weighing the Reference Boxes.** Recall that the marginal gain of our proposed diversity function rewards covering the reference boxes with the chosen set of boxes. Instead of weighing all reference boxes equally, we found it important to weigh different reference boxes differently. The exact form the weighting rule is provided in the supplement. In our experiments, we present results with and without such a weighting to show impact of our proposed scheme.

## 5.1 Accuracy of Object Proposals

In this section, we explore the performance of our proposed method in comparison to relevant object proposal generators. For the two PASCAL datasets, we perform cross validation on 2510 validation images of PASCAL VOC 2007 for the best parameter $\lambda$, then report accuracies on 4952 test images of PASCAL VOC 2007 and 5823 validation images of PASCAL VOC 2012. The MS COCO dataset is much larger, so we randomly select a subset of 5000 training images for tuning $\lambda$, and test on complete validation dataset with 40138 images.

We use 1000 top ranked selective search windows [39] as reference boxes. In a manner similar to [23], we chose a different $\lambda_M$ for $M = 100, 200, 400, 600, 800, 1000$ proposals. We compare our approach with several baselines: 1) $\lambda = \infty$, which essentially involves re-ranking selective search windows by considering their ability to coverage other boxes. 2) Three variants of EdgeBoxes [41] at IoU $= 0.5, 0.7$ and $0.9$, and corresponding three variants without affinities in (3). 3) Selective Search: compute multiple hierarchical segments via grouping superpixels and placing bounding boxes around them. 4) SS-EB: use EdgeBoxesScore to re-rank Selective Search windows.

Fig. 4 shows that our approach at $\lambda = \infty$ and validation-tuned $\lambda$ both outperform all baselines. At $M = 25, 100$, and $500$, our approach is $20\%, 11\%$, and $3\%$ better than Selective Search and $14\%, 10\%$, and $6\%$ better than EdgeBoxes70, respectively.

## 5.2 Ablation Studies.

We now study the performance of our system under different components and parameter settings.

**Effect of $\lambda$ and Reference Boxes.** We test performance of our approach as a function of $\lambda$ using reference boxes from different object proposal generators (all reported at $M$=200 on PASCAL VOC 2012). Our reference box generators are: 1) Selective Search [39]; 2) MCG [2]; 3) CPMC [7]; 4) EdgeBoxes [41] at IoU $= 0.7$; 5) Objectness [1]; and 6) Uniform-sampling [20]: *i.e.* uniformly sample the bounding box center position, square root area and log aspect ratio.

Table 1 shows the performance of SubmodBoxes when used with these different reference box generators. Our approach shows improvement (over corresponding method) for *all reference boxes*. Our approach outperforms the current state of art MCG by $2\%$ and Selective Search by $5\%$. This is significantly larger than previous improvements reported in the literature.

Fig. 5a shows more fine-grained behavior as $\lambda$ is varied. At $\lambda = 0$ all methods produce the same (highest weighted) box $M$ times. At $\lambda = \infty$, they all perform a reranking of the reference set of boxes. In nearly all curves, there is a peak at some intermediate setting of $\lambda$. The only exception is EdgeBoxes, which is expected since it is being used in both the relevance and diversity terms.

**Effect of No. B&B Steps.** We analyze the convergence trends of B&B. Fig. 5b shows that both the optimization objective function value and the mABO increase with the number of B&B iterations.

|  | Selective-Search | MCG | EB | CPMC | Objectness | Uniform-sampling |
|---|---|---|---|---|---|---|
| $\lambda \approx 0.4$, weighting | **0.7342** | 0.7377 | **0.6747** | 0.7125 | **0.6131** | **0.5937** |
| $\lambda \approx 0.4$, without weighting | 0.5697 | 0.5042 | 0.6350 | 0.5681 | 0.6220 | 0.5136 |
| $\lambda = 10$, weighting | 0.7233 | **0.7417** | 0.6467 | **0.7130** | 0.5006 | 0.5478 |
| $\lambda = 10$, without weighting | 0.5844 | 0.5534 | 0.6232 | 0.5849 | 0.5920 | 0.5115 |
| $\lambda = \infty$, weighting | 0.7222 | 0.7409 | 0.6558 | 0.7116 | 0.4980 | 0.5453 |
| Original method | 0.6817 | 0.7206 | 0.6755 | 0.7032 | 0.6038 | 0.5295 |

Table 1: Comparison with/without weighting scheme (rows) with different reference boxes (columns). 'Original method' row shows performance of directly using object proposals from these proposal generators. '$\approx$' means we report the best performance from $\lambda = 0.3, 0.4$ and $0.5$ considering the peak occurs at different $\lambda$ for different object proposal generators.

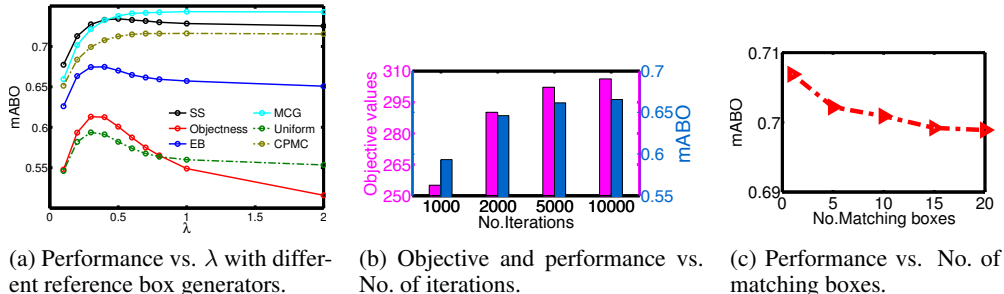

(a) Performance vs. $\lambda$ with different reference box generators.

(b) Objective and performance vs. No. of iterations.

(c) Performance vs. No. of matching boxes.

Figure 5: Experiments on different parameter settings.

**Effect of No. of Matching Boxes.** Instead of allowing the chosen boxes to cover exactly one reference box, we analyze the effect of matching top-k reference boxes. Fig. 5c shows that the performance decreases monotonically bit as more matches are allowed.

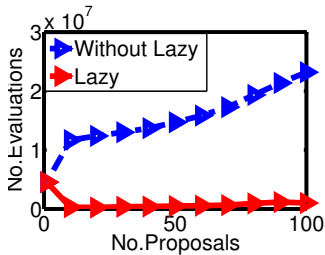

Figure 6: Comparison of the number of B&B iterations of our Lazy Greedy generalization and independent B&B runs.

**Speed up via Lazy Greedy.** Fig. 6 compares the number of B&B iterations required with and without our proposed Lazy Greedy generalization (averaged over 100 randomly chosen images) – we can see that Lazy Greedy significantly reduces the number of B&B iterations required. The cost of each B&B evaluation is nearly the same, so the iteration speed-up is directly proportional to time speed-up.

## 6 Conclusions

To summarize, we formally studied the search for a set of diverse bounding boxes as an optimization problem and provided theoretical justification for greedy and heuristic approaches used in prior work. The key challenge of this problem is the large search space. Thus, we proposed a generalization of Minoux's 'lazy greedy' on B&B tree to speed up classical greedy. We tested our formulation on three datasets of object detection: PASCAL VOC 2007, PASCAL 2012 and Microsoft COCO. Results show that our formulation outperforms all baselines with a novel diversity measure.

**Acknowledgements.** This work was partially supported by a National Science Foundation CAREER award, an Army Research Office YIP award, an Office of Naval Research grant, an AWS in Education Research Grant, and GPU support by NVIDIA. The views and conclusions contained herein are those of the authors and should not be interpreted as necessarily representing the official policies or endorsements, either expressed or implied, of the U.S. Government or any sponsor.

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
