[Reviews · NeurIPS 2015]

Submitted by Assigned_Reviewer_1

The authors propose combining a branch-and-bound technique with lazy greedy to speed up submodular maximization. The idea to cleverly search for the best marginal gain appeared at last year's NIPS [28], although the setting is different as the authors note. What worries me, is that their technique does not seem easy to lift to other problems, and it seems too specialized for the proposed functions. Both the modular term and the diversity term have to be chosen so that the whole set of sets we consider can be easily upper bounded. For example, the authors simplify the modular score and also use a very simple diversity function to be able to efficiently compute upper bounds.

Some additional comments:

a) Regarding NMS, can't you use say 1/0 instead of 0/-infty? Then, the traditional NMS algorithm can be seen as greedily maximizing this marginal gains, which are never negative.

b) It would have been nice to show the time-quality trade-off you've made by using the unsimplified EdgeBoxes scores, or a more complicated diversity function that's hard to upper bound using your strategy.
Summary: A well-written paper speeding up submodular maximization. My main concern is that the technique might be too tightly bound with the application at hand. While this paper will be probably better appreciated at a vision conference, it might also find audience at NIPS.

Submitted by Assigned_Reviewer_2

SUMMARY

The paper introduces a principled approach to generate a diverse set of relevant object proposals. This is formulated as optimizing a function F defined on sets of bounding boxes. This function is defined as a sum of a relevance term (which encourages individual bounding boxes to be likely to contain an object) and a diversity term. For optimization they commit to a greedy approach which iteratively adds the most promising bounding box, until the desired number of proposals was generated. The greedy procedure was motivated by existing theoretical guarantees on the obtained solution when F is submodular and monotone. Even under this greedy procedure, the computational cost would be prohibitive when using the naive approach of scanning all possible bounding boxes in order to find the best one. To this end, a Branch and Bound (B&B) procedure is employed. Furthermore, the efficiency of the B&B procedure is improved by sharing information between successive steps of the greedy procedure, generalizing ideas from Minoux's Lazy Greedy procedure.

COMMENTS

On the positive side, the paper introduces an extensible framework for object proposal generation, which is shown to encompass some previous heuristics like NMS and the one used in the ESS [22] method. The branch and bound procedure is interesting and makes it feasible to (greedily) search for the solution despite the very large number of candidate bounding boxes considered at each step. This involved constructing an upper bound for the marginal gain of F given a set of candidate bounding boxes. One technical novelty consists of generalizing Minoux's Lazy Greedy procedure in order to leverage information from the B&B tree constructed in the previous step of the greedy procedure. This procedure was shown to perform well in practice. Furthermore, the paper is clearly written and well organized.

There appears to be an issue with the monotonicity and submodularity proofs presented in the supplementary material. In brief, this is related to the fact that these concepts are defined on functions over sets (not over lists), and the theoretical guarantees mentioned refer to this situation. The proposed method is defined in terms of a list (lines 179-181), but the diversity function should be defined on the corresponding unordered sets (not lists) in order for the theoretical guarantees to be available. However, the definition in eqs. 7a, 7b can lead to different values of the diversity function D, for two lists which differ only in the order of the proposals (e.g. for different orderings, the same proposal might be attached to different reference bounding boxes, leading to different values when summing up the contributions from all proposals) -- therefore the elements of the underlying set cannot be considered in an arbitrary order when evaluating D, so the proof has to be adjusted. How do these aspects influence the theoretical claims regarding the performance of the greedy procedure?

The results in figure 4 show a comparison between the proposed method and variants of Edge Boxes and Selective Search. It would be best to compare to additional state-of-the-art methods, such as MCG and CPMC (which are present in table 1). The results in table 1 are obtained under the regime of a small number of proposals (200 per image, but methods like MCG and CPMC can be tuned for as many as thousands of proposals if not more), which is not frequently used in practice. For such reasons, the impact of the proposed method is not clear from an empirical perspective.

The extension of the greedy procedure was shown in fig. 6 to lead to evaluating significantly fewer nodes of the B&B graph. It would be interesting to also show the effect in terms of execution time.

Minor comments: there is a double caption for fig. 3.
Summary: The paper presents a principled framework to generate object proposals and introduces algorithmic contributions to make it feasible. The final results are not entirely convincing and there are some issues with the theoretical guarantees claimed, which should be clarified prior to publication.

Submitted by Assigned_Reviewer_3

Summary: This paper introduces a theoretical justification of greedy algorithms for detecting multiple objects in an image using submodularity, shows how branch-and-bound can be used for each detection step, introduces a way of speeding up branch-and-bound by caching results from each detection step, show how to apply this successfully for generating object proposals, and shows how commonly used existing techniques for detection can be understood within this theoretical framework.

Quality: This is a fascinating paper with many interesting theoretical components and fairly convincing experiments.

I would probably view it as a top 5% of accepted NIPS papers if I hadn't come across the uncited paper "Branch and bound strategies for non-maximal suppression in object detection" from Blaschko et al., which to my understanding contains several related components.

Clarity: The paper is well written and clear

Originality:

To my understanding, the paper Blaschko et al. "Branch and Bound Strategies for Non-maximal Suppression in Object Detection" has several related components to this submission 1) the idea of using submodularity to obtain approximation guarantees of greedy multi-object detection algorithms, 2) using branch-and-bound search for each iteration of greedy search, and 3) re-using priority queues between each round of branch-and-bound.

To be fair, I'm uncertain as to whether or not the Minoux' lazy greedy algorithm presented in the paper is different with respect to this 3rd point.

This submission also contains details to apply the method as an object proposal method, and an interesting summary of how commonly used techniques in object detection fit into this theoretical framework.

At the same time, I think that this paper would have to be re-written to put itself in perspective with the Blaschko paper.

Significance: It is clearly interesting and worthwhile to have a better theoretical understanding of object detection algorithms, and also significant if this theoretical understanding can be used to improve detection algorithms in practice.
Summary: This is a well written paper that includes interesting theoretical analysis and several intriguing components.

However, I believe this paper missed a major citation that might subsume the main contributions of this submission related to theoretical analysis using submodularity, the use of branch-and-bound, and a method to cache priority queues during branch-and-bound.

Submitted by Assigned_Reviewer_4

The paper is very well written and the authors do a good job at explaining several concepts involving branch-and-bound algorithm.

The paper provides a nice and principled generalization to the Efficient subwindow search algorithm. The extension of branch and bound process for generating multiple bounding boxes is also pretty good.

The paper specifically accounts for diversity measure, which leads to a good coverage over the entire image. Diversity has been used in the past during re-ranking the proposals (e.g. CPMC, objectness etc), but using diversity during the generation process itself is an interesting approach.

A nice extension to Greedy-submodular maximization over structured output spaces is also presented, which can be very beneficial for general subset selection problems over structured spaces.

The results are pretty impressive and beats many state-of-the-art methods which shows the importance of diversity for generating higher recalls with smaller number of proposals.

Weakness: The proposed diversity measure had to rely on a set of reference bounding boxes for efficient computation of upper bounds. The authors commit themselves to the weaknesses of existing proposal methods (selective search in this case).

It's puzzling to me that a simple re-ranking of selective search (\lambda=inf), performs almost the same as the weighted lambda. Does this indicate that the performance is very tightly coupled with the quality of reference boxes? Adding another result with different set of reference boxes may help in understanding this better.
Summary: The paper makes several theoretical contributions and shows the value of incorporating diversity in object proposal methods. The experimental results are pretty impressive as well.

Author Feedback
Author rebuttal: We thank the reviewers for their time and feedback.

We are very pleased that reviewers found our work making principled theoretical contributions (R1,2,3), the paper well-written (R1,3), and the experiments convincing/impressive (R2,3).

We address the major comments below, and will incorporate all feedback in the final version.

1. R2: Novelty over Blaschko EMMCVPR11 [B11]

Thank you for bringing this to our attention. This was published at a non-traditional CV/ML venue and we were unaware. Here are the major differences:

- The focus of [B11] is the use of B&B for NMS in object detection. In that sense, it is similar to the ESS heuristic [22], which also used B&B for finding multiple objects. Our work presents a general framework for bounding-box subset-selection based on submodular maximization and shows that a number of previous works (NMS and [22]) are special cases of this framework.

- Most importantly, although [B11] justify their use of greedy from the perspective of submodular maximization, their claim about approximation guarantees of their approach (page5, para2) is simply incorrect.
Greedy has (1-1/e)-approximation guarantee for (cardinality constrained) /monotone/ submodular maximization. The objective in [B11] (Eq 1) is submodular but not monotone. As we explain in our paper (L249-258) and [B11] point out (page 6), NMS has negative marginal gains. Unfortunately, [B11] does not seem to have realized that this means greedy no longer has guarantees and different techniques are needed [3,12].
[Note: [B11] points to Barinova et al CVPR10/PAMI12 for guarantee of greedy, but Barinova et al achieve their guarantees from the "facility localization" problem, not submodular maximization. The objective in [B11] is not an instance of facility localization either.]

This observation about NMS was one of the reasons we proposed a coverage-based diversity function that is submodular and monotone.
We believe this mistake in existing literature makes it more critical that our work be published.

- [B11] does reuse the priority queue, but this is similar to the way [22] reuses the priority queue. Theoretically, neither work makes connections with Minoux's lazy greedy (which leads to a subtle but important difference in the way the re-use happens; the upper-bound updates are interleaved with B&B in lazy greedy). Empirically, the [B11] heuristic for reusing priority queue is slower. We experimentally compared and found our approach is 28% faster than B[11] on average.

We add these to the final version.

2. R1: lists vs sets
We apologize for using both sets and lists in our presentation (we thought sets would be intuitive early in the paper). To be clear, our work falls under submodular list prediction. As described in L179, the generalization from sets to lists allows reasoning about an ordering of the items chosen and (potentially) repeated entries in the list.

All guarantees about greedy remain the same in this generalization. More details about list prediction can be found in Streeter & Golovin NIPS09, [8], Ross et al. ICML13. We will change the final version entirely to lists to avoid confusion.

3. R1: Comparison to other methods (MCG and CPMC)
We broke the various comparisons down into separate figures to avoid clutter. Fig 4 compares our approach to the two "ingredients" in our work - EdgeBoxes and Selective Search. As R1 notes, Table 1 does contain comparisons to numerous other methods (including MCG and CPMC).

4. R1: #proposals 200 too few
First, note that our approach can easily work at higher #proposals (cross-val would presumably end up picking a small diversity parameter lambda). Second, note that recent works [Krahenbuhl & Koltun CVPR15, Ren et al. arxiv:1506.01497] are already operating at 650 & 300 proposals (not M~=1000s). As better proposal methods develop, M will decrease further, making our reported regime even more relevant.

5. R3: lambda=inf performance indicates choice references boxes important
Note that Table 1 shows 1-11% improvement of val-tuned-labmda vs lambda=inf, depending on the reference box.
But we agree - the choice of the reference box is important. This is the reason we performed an extensive evaluation with 6 different reference boxes (SS, MCG, EdgeBoxes, CPMC, Objectness, Uniform Sampling) in Fig 5b and Table 1. Our approach shows improvement (over corresponding ref box method) with /all/ reference boxes. Best performing reference generators are SS and MCG.

6. R4: NMS: 1/0 vs 0/-infty
No, unfortunately 1/0 does not work. In order to enforce a hard constraint of suppressing overlapping boxes, an infinite penalty is needed (otherwise an overlapping box with sufficiently high relevance score could be selected). This is similar to Lagrange variables in duality theory.

7. R1: Fig 6 wrt time
The cost of each B&B evaluation is nearly constant, so the x-axis is proportional to time. We will add the time plot.